# Synthesis of Novel *ent*-Kaurane-Type Diterpenoid Derivatives Effective for Highly Aggressive Tumor Cells

**DOI:** 10.3390/molecules23123216

**Published:** 2018-12-05

**Authors:** Yu Hu, Xiao-Nian Li, Ze-Jin Ma, Pema-Tenzin Puno, Yong Zhao, Yan Zhao, Ye-Zhi Xiao, Wei Zhang, Jing-Ping Liu

**Affiliations:** 1Department of Chemistry, Yunnan Normal University, Kunming 650092, China; xpcxyy@sina.com (Y.H.); mazejinmzj@sina.com (Z.-J.M.); zhaooy@126.com (Y.Z.); zhaooyann@163.com (Y.Z.); xyzlmm@sina.com (Y.-Z.X.); royzhangwei@sina.com (W.Z.); 2State Key Laboratory of Phytochemistry and Plant Resources in West China, Kunming Institute of Botany, Kunming 650201, China; lixiaonian@mail.kib.ac.cn

**Keywords:** *ent*-kaurane-type diterpenoid derivatives, antitumor, synthesis

## Abstract

We have designed and synthesized 6 *ent*-Kaurane-type diterpenoid derivatives containing α,β-unsaturated ketone moieties. In vitro, activity was evaluated against three human tumor cell lines and a rat myogenic cell line (HepG2, NSCLC-H292, SNU-1040, L6) by MTT assay. All the tested compounds exhibited comparable or higher activity than DDP and eriocalyxin B. Compounds **16**, **17** and **18** are promising anti-tumor leads due to their cytotoxic potencies and higher selectivity, with SI values of 161.06, 47.80 and 128.20, respectively.

## 1. Introduction

Highly aggressive tumors are a major cause of death worldwide, accounting for 8.8 million deaths each year, and are estimated to cause 9.6 million deaths in 2018. Globally, about 1 in 6 deaths is due to cancer. Approximately 70% of deaths from cancer occur in low- and middle-income countries, according to the World Health Organization (WHO) [1].

Plants are normally an important source of new drugs [2]. In recent years, *Isodon* genus members have been promising leads due to their extensive pharmacological and physiological effects, such as inhibition of hepatitis viral replication [3] and bacterial infections of the lung or gut [4], as well as its antimalaria [5], antiinflammatory [6], and anti-tumor properties [7]. Hitherto, over 1200 *ent*-Kaurane-type diterpenoids with highly compact polycyclic ring systems have been isolated from *Isodon* genus, notably the compounds oridonin [8]. Natural Diterpenoid Isoferritin A (IsoA) [9], longikaurin [10], xerophilusins [11] (Figure 1, **1**–**4**), which have been studied for anti-tumor activity. A common structural unit presented by these diterpenoid compounds is the functionalized bridged C/D α,β-unsaturated ketone moiety. Unfortunately, the use of these natural diterpenoids for anti-tumor was hampered by its moderate potency and complex oxygenated diterpenoid scaffold. Therefore, it is very necessary to find compounds with simple structure and stronger activity.

In the preceding letter we reported the synthesis and biological evaluation of a series of eriocalyxin B derivatives (Figure 2, **5**–**7**) and found that eriocalyxin B derivatives bearing one or two α,β-unsaturated ketone units are potent antitumor agents [12]. It is noteworthy that the 7-hydroxyl group and the hydrogen bond between 6-hydroxyl group and 15-carbonyl group are not necessary for antitumor activity.

## 2. Results and Discussion

### 2.1. Chemistry

Herein we report the synthesis of ent-Kaurane-type diterpenoid derivatives containing two α,β-unsaturated ketone moieties and their in vitro cytotoxic properties against three human tumor cell lines. Our synthesis began with reacting isobutyraldehyde with methyl vinyl ketone (MVK) in the presence of sulfuric acid which produced 4,4-dimethylcyclohex-2-enone (compound **8**). Treatment of commercially available 1,3-cyclohexan-dione with ethanol in a reflux of benzene in the presence of p-toluene sulfonic acid resulted in 3-ethoxy-cyclohex-2-enone (compound **9**) (86%). Compound **9** was treated sequentially with NaH and diethyl carbonate to directly produce β-ketoester. Treatment of β-ketoester with NaH and allyl bromide produced compound **10** (81%). The resulting compound **10** was reduced using NaBH_4_ followed by acidification to produce compound **11**. Following the well- established Ma protocol [13], the key intermediate **12** (52%) was obtained (Scheme 1).

Deprotonation of **8** with LiHMDS followed by a reaction with compound **12** led to compound **13** as the major product, along with isomer **14** in 82% combined yield (dr = 60:40). The structures of **13** and **14** were confirmed by NMR spectral and X-ray crystallographic analysis (Figure 3). Next, compound **13** was subjected to allylic oxidation with selenium dioxide (SeO_2_, CH_2_Cl_2_, t-BuOOH) and IBX oxidation to form compounds **15** and **16**, respectively. Compound **17** (or **18**) was provided from the diastereoisomer **14** under the same conditions as **15** (or **16**) (Scheme 2).

### 2.2. Biological Evaluation

The cytotoxic properties of all newly synthesized compounds were evaluated in vitro against three human tumor cell lines ((HepG2, NSCLC-H292, SNU-1040) and one rat myotubes cell line (L6) by MTT assay. Cisplatin (DDP) and eriocalycin B (**5**) were used as reference drugs. The results are summarized in Table 1. (IC_50_ value is defined as the concentrations corresponding to 50% growth inhibition). Remarkably, all the tested compounds exhibited comparable or even higher activity than their natural references eriocalyxin B and DDP. Although results are preliminary, we found a clear structure-activity relationship: the α,β-unsaturated ketone moiety is necessary for potency. Particularly for compound **16** (IC_50_ = 0.33; 1.69; 2.44 μM) and **18** (IC_50_ = 0.56; 1.35; 3.01 μM), which display two α,β-unsaturated ketone moieties, these had stronger in vitro potencies and cytotoxic activities to DDP. To obtain insight into the cytotoxic potential of these new compounds on normal cells, the effect of compounds **13**, **14**, **15**, **16**, **17**, **18** and DDP (cisplatin) were evaluated in human liver carcinoma cell (HepG2) and rat myotubes (L6) cells. All the tested analogues exhibited higher selectivity than DDP. Especially, compound **16**, which had more potency and selectivity than DDP and eriocalyxin B, with an SI (selective index, IC_50_ of normal cells/IC_50_ of tumor cells) value of 161.06. Interestingly, the two epimers **16**, **18** showed no significant difference in their anticancer activities.

## 3. Materials and Methods

### 3.1. Chemistry

All regents were used as purchased from Energy Chemical (Shanghai, China), Tansoole (Shanghai, China), Acros (Beijing, China) and Aldrich (Shanghai, China) without further purification. Anhydrous CH_2_Cl_2_ was freshly distilled from CaH_2_, and THF was dried by Na. For reactions carried out under Ar, at least three times. The course of reaction was monitored by TLC. ^1^H NMR and ^13^C NMR spectra were recorded at ambient temperature with a Bruker Avance 300 (300 MHz for ^1^H and 75.5 MHz for 13C; Bruker, Germany) instrument in which TMS was used as internal standard for all measurements. MS data were recorded by using a VG Auto spec-3000 spectrometer or a Finnigan MAT 90 instrument (Bremen, Germany). IR spectra were measured as KBr pellets by using a Bio-Rad FTS-135 spectrometer (California, CA, USA). All melting points are uncorrected. CC was performed by using silica gel (100–200 mesh; Qingdao Marine Chemical, Shandong, China).

*4,4-Dimethylcyclohex-2-enone* (**8**): To a solution of isobutyraldehyde (3 g, 41.6 mmol) and methyl vinyl ketone (2.91 g, 41.6 mmol) was added 1% mmol conc. H_2_SO_4_. The resulting mixture was stirred with a Dean-Stark apparatus in refluxing benzene (20 mL) for 5 h. The reaction mixture was neutralized by sat. NaHCO_3_ to pH = 7. Organic layer was separated and water was extracted with EtoAc (3 × 10 mL). The organic phases were combined, dried over MgSO_4_, and concentrated on a rotary evaporator. The residue was purified by flash chromatography (silica, ethyl acetate:petroleum = 1:10) to give compound 8 (92%) as pale yellow liquid: ^1^H NMR (400 MHz, CDCl_3_) δ = 6.67(d, *J* = 6 Hz, 1H), 5.82 (d, *J* = 6 Hz, 1H), 2.45 (t, *J* = 6 Hz, 2H), 1.87 (t, *J* = 6 Hz, 2H), 1.17 (s, 6H) ppm; ^13^C NMR (100 M Hz, CDCl_3_) δ = 199.48, 159.79, 126.77, 36.03, 34.34, 32.77, 27.64 ppm; IR (KBr):υ_max_ = 3441.89, 2966.23, 2926.56, 1638.51 cm^−1^; MS (ESI): *m*/*z*: 125; HRMS (ESI) Calcd. for [(C_8_H_12_O) + H]^+^: 125.0966; found: 125.0961 [M + H]^+^.

*3-Ethoxycyclohex-2-enone* (**9**): To a solution of cyclohexane-1,3-dione (1.12 g, 10 mmol) in dry ethanol was added 1% mmol TsOH. The resulting mixture was refluxed for overnight. Then the solvent was removed in vacuo and the residue was purified by flash column chromatography (eluting with EtOAc:petroleum = 1:20–1:10) to give compound **9** (86%) as colorless liquid: ^1^H NMR (300 MHz, CDCl_3_) δ = 4.74 (s, 1H), 3.36 (q, *J* = 6.9 Hz, 2H), 1.85 (t, *J* = 6.0 Hz, 2H), 1.74 (t, *J* = 6.0 Hz, 2H), 1.11 (p, *J* = 6.0 Hz, 2H), 0.80 (t, *J* = 6.9 Hz, 3H) ppm; ^13^C NMR (75.5 MHz, CDCl_3_) δ = 197.496, 176.357, 101.367, 62.996, 35.635, 27.897, 20.201, 12.970 ppm; IR (KBr):υ_max_ = 2944.51, 1591.52 cm^−1^; MS (ESI): *m*/*z*: 141; HRMS (ESI) Calcd. for [(C_8_H_12_O_2_) + H]^+^: 141.0916; found: 141.0906 [M + H]^+^.

*Ethyl 1-allyl-4-ethoxy-2-oxocyclohex-3-enecarboxylate* (**10**): NaH (0.6 g, 15 mmol, 60% in mineral oil) was suspended in diethyl carbonate (10 mL) at 100 °C. A solution of **9** (1.4 g, 10 mmol) in diethyl carbonate (2 mL) was added to the suspension by syringe and stirring was continued for 5 h at 100 °C. Upon cooling to ambient temperature, the reaction mixture was washed with brine (2 × 10mL) and evaporated in vacuo without further purification. To a solution of crude product in dry THF was added NaH (0.4 g, 10 mmol, 60% in mineral oil) at room temperature. After 30 min, the mixture was added allyl bromide (1.2 g, 10 mmol) and refluxed for 3 h. The reaction mixture was poured into ice-water and extracted with ether. The ether layer was washed with water and dried over Na_2_SO_4_. Removal of solvent followed by CC gave compound **10** (81%) as colorless liquid: ^1^H NMR (300 MHz, CDCl_3_) δ = 5.77 (m, 1H), 5.35 (s, 1H), 5.09 (m, 2H), 4.17 (q, *J* = 6.9 Hz, 2H), 3.92 (q, *J* = 6.9 Hz, 2H), 2.74–2.32 (m, 5H), 1.97–1.91 (m, 1H), 1.19 (t, *J* = 6.9 Hz, 3H), 1.05 (t, *J* = 6.9 Hz, 3H) ppm; ^13^C NMR (75.5 MHz, CDCl_3_) δ = 194.98, 176.59, 171.13, 133.50, 118.42, 101.65, 64.24, 60.98, 55.33, 38.38, 37.95, 26.10, 13.93 ppm; IR (KBr):υ_max_ = 2979.73, 2937.66, 1725.23, 1655.39, 1605.02, 1186.77 cm^−1^; MS (ESI): *m*/*z*: 275; HRMS (ESI) Calcd. for [(C_14_H_20_O_4_) + Na]^+^: 275.1259; found: 275.1254 [M + Na]^+^.

*Ethyl 1-allyl-4-oxocyclohex-2-enecarboxylate* (**11**): To a solution of 10 (1 g, 3.97 mmol) in MeOH (5 mL) was added CeCl_3_·7H_2_O (1.48 g, 3.97 mmol). After 10 min of stirring, the reaction mixture was added NaBH_4_ (0.3 g, 7.94 mmol) about for 30 min. Then the mixture was acidified to pH 6 by the dropwise addition of concentrated HCl, and extracted with ethyl acetate and washed with brine. The combined organic layers were dried over Na_2_SO_4_ and the solvent was removed under reduced pressure. Purification of the residue by column chromatograph afford compound **11** (84%) as pale yellow liquid: ^1^H NMR (300 MHz, CDCl_3_) δ = 6.88(d, *J* = 10.2 Hz, 1H), 5.97 (d, *J* = 10.2 Hz, 1H), 5.66 (m, 1H), 5.10 (m, 2H), 4.15 (q, *J* = 7.2 Hz, 2H), 2.52–2.34 (m, 5H), 1.98–1.93 (m, 1H), 1.24 (t, *J* = 7.2 Hz, 3H) ppm; ^13^C NMR (75.5 MHz, CDCl_3_) δ = 198.498, 172.728, 150.539, 131.920, 129.227, 119.638, 61.435, 47.502, 42.884, 34.525, 30.238, 14.184 ppm; IR (KBr):υ_max_ = 2978.78, 2958.27, 1728.05, 1638.95, 1614.26 cm^−1^; MS (ESI): *m*/*z*: 209; HRMS (ESI) Calcd. for [(C_12_H_16_O_3_) + H]^+^: 209.1178; found: 209.1169 [M + H]^+^.

*Ethyl 6-methylene-4-oxobicyclo[3.2.1]oct-2-ene-1-carboxylate* (**12**): To a solution of 11 (6 g, 28.85 mmol, 1.0 eq) in dry THF (200 mL) under Ar at –60 °C was added Et_3_N (11.6 mL, 115.4 mmol, 4.0 eq). The mixture was stirred for 20 min at this temperature, and then TBSOTf (19.11 g, 72.13 mmol, 2.5 eq) was added dropwise. After full conversion was observed by TLC, the reaction mixture was quenched with saturated NaHCO_3_ solution and extracted with EtOAc. The combined organic layers were dried over Na_2_SO_4_ and the solvent was removed under reduced pressure. The residue was passed through a short column of silica gel to provide the corresponding silyl enol ether as a colorless oil. The obtained silyl enol ether was dissolved in DMSO (200 mL) and Pd(OAc)_2_ (650 mg, 2.89 mmol, 0.1 eq) was added, and the resulting solution was stirred under oxygen atmosphere (balloon with O_2_) at 40 °C for 6 h. The reaction was quenched with water, the mixture was extracted with Et_2_O, and the combined organic layers were washed with brine, dried over Na_2_SO_4_, filtered, and concentrated under reduced pressure. The residue was purified by flash column chromatography to afford compound 12 (40%) as pale yellow liquid: ^1^H NMR (300 MHz, CDCl_3_) δ = 7.43 (d, *J* = 9.9 Hz, 1H), 5.81 (d, *J* = 9.9 Hz, 1H), 5.26 (s, 1H), 5.05 (s, 1H), 4.19 (q, *J* = 7.2 Hz, 2H), 3.47 (br., 1H), 2.86 (d, *J* = 95.4 Hz, 1H), 2.52 (d, *J* = 95.4 Hz, 1H), 2.25 (m, 2H), 1.25 (t, *J* = 7.2 Hz, 3H) ppm; ^13^C NMR (75.5 MHz, CDCl_3_) δ = 197.515, 172.761, 152.359, 143.136, 126.641, 113.126, 61.584, 57.960, 51.617, 43.207, 41.109, 14.120 ppm; IR (KBr):υ_max_ = 2979.98, 1729.06, 1685.40, 1280.90, 1222.04, 1193.09 cm^−1^; MS (ESI): *m*/*z*: 207; HRMS (ESI) Calcd. for [(C_12_H_14_O_3_) + H]^+^: 207.1021; found: 207.1015 [M + H]^+^.

#### 3.1.1. General Method for Synthesis of **13** and **14**

To a stirred solution of **8** (1 eq) and **12** (1 eq) in dry THF under Ar at −78 °C; was added LiHMDS (2 eq, 1.3 M in THF) over a period of 5 min and stirring at −78 °C was continued for 45 min. The resulting solution was warmed to room temperature, quenched with saturated NH_4_Cl and extracted with Et_2_O. The combined organic layers were dried over Na_2_SO_4_ and the solvent was removed under reduced pressure. The residue was purified by flash column chromatography to provide **13** and **14**.

Compound **13**: The compound is obtained as crystal; m.p.: 112.0–114.6 °C; ^1^H NMR (300 MHz, CDCl_3_) δ = 6.56 (d, *J* = 6.0 Hz, 1H), 5.80 (d, *J* = 6.0 Hz, 1H), 5.10 (s, 1H), 5.01 (s, 1H), 4.15 (q, *J* = 6.0 Hz, 2H), 3.50 (m, 1H), 3.28 (s, 1H), 2.78 (s, 2H), 2.74–2.65 (m, 2H), 2.28 (m, 1H), 1.92–1.66 (m, 4H), 1.29 (t, *J* = 6.0 Hz, 3H), 1.26 (s, 3H), 1.25 (s, 3H) ppm; ^13^C NMR (75.5 MHz, CDCl_3_) δ = 208.93, 198.35, 174.52, 158.35, 145.59, 126.50, 109.91, 61.00, 58.73, 52.97, 46.11, 43.09, 38.80, 38.56, 36.41, 36.11, 33.70, 30.64, 24.95, 14.21 ppm; IR (KBr):υ_max_ = 2962.26, 2901.21, 1722.91, 1710.24, 1672.31, 1229.31, 1194.50 cm^−1^; MS (ESI): *m*/*z*: 353; HRMS (ESI) Calcd. for [(C_20_H_26_O_4_) + Na]^+^: 353.1729; found: 353.1724 [M + Na]^+^.

Compound **14**: The compound is obtained as crystal; m.p.:103.2–105.8 °C; ^1^H NMR (300 MHz, CDCl_3_) δ = 6.46(d, *J* = 6.0 Hz, 1H), 5.68 (d, *J* = 6.0 Hz, 1H), 5.04 (s, 1H), 4.88 (s,1H), 4.11 (q, *J* = 6.0 Hz, 2H), 3.30 (s, 1H), 2.65–2.39 (m, 6H),1.96 (m, 1H), 1.78–1.22 (m, 3H), 1.01 (t, *J* = 6.0 Hz, 3H), 0.88 (s, 6H) ppm; ^13^C NMR (75.5 MHz, CDCl_3_) δ = 206.74, 199.50, 175.14, 157.76, 146.00, 127.27, 109.41, 60.96, 59.66, 52.69, 45.89, 44.46, 43.35, 42.79, 37.86, 35.31, 34.03, 30.39, 24.74, 14.26 ppm; IR (KBr):υ_max_ = 2958.90, 1716.70, 1673.18, 1538.19, 1234.84, 1215.98 cm^−1^; MS (ESI): *m*/*z*: 353; HRMS (ESI) Calcd. for [(C_20_H_26_O_4_) + Na]^+^: 353.1729; found: 353.1732 [M + Na]^+^.

#### 3.1.2. General Method for Synthesis of **15** and **17**

To a stirred solution of **13** (1.0 eq) in DCM, SeO_2_ (0.9 eq) was added in one portion, and a solution of t-BuOOH (5.5 M in decane, 3 eq) was added dropwise. The solution was vigorously stirred at room temperature overnight, and then the mixture was filtered through a short pad of silica gel washed with EtOAc. The filtrate was concentrated under vacuum and purified by flash column chromatography to provide allylic alcohol **15**.

Compound **15**: The compound is obtained as pale yellow liquid; ^1^H NMR (300 MHz, CDCl_3_) δ = 6.52 (d, *J* = 6 Hz, 1H), 5.75 (d, *J* = 6 Hz, 1H), 5.40 (s, 1H), 5.30 (s, 1H), 4.58 (s, 1H), 4.16 (q, *J* = 6 Hz, 2H), 3.32 (m, 2H), 2.57 (m, 3H), 2.17–1.69 (m, 5H), 1.21 (t, *J* = 6 Hz, 3H), 1.08 (s, 6H) ppm; ^13^C NMR (75.5 MHz, CDCl_3_) δ = 208.33, 198.08, 172.65, 158.66, 149.08, 126.53, 115.31, 79.98, 61.18, 59.47, 56.68, 46.35, 37.91, 36.43, 35.20, 33.54, 32.64, 30.62, 25.07, 14.28 ppm; IR (KBr):υ_max_ = 3451.12, 2961.46, 2927.59, 2871.68, 1714.49, 1669.43, 1280.80, 1223.47, 1199.50, 1059.84, 1037.64 cm^−1^; MS (ESI): *m*/*z*: 369; HRMS (ESI) Calcd. for [(C_20_H_26_O_5_) + Na]^+^: 369.1678; found: 369.1677 [M + Na]^+^.

Compound **17**: The compound is obtained as pale yellow liquid; ^1^H NMR (300 MHz, CDCl_3_) δ = 6.48 (d, *J* = 9 Hz, 1H), 5.68 (d, *J* = 9 Hz, 1H), 5.36 (s, 1H), 5.31 (s, 1H), 4.46 (s, 1H), 4.15 (q, *J* = 6 Hz, 2H), 3.37 (m, 1H), 2.45–2.39 (m, 5H), 1.98 (m, 2H), 1.78 (m, 1H), 1.68 (m, 1H), 1.24 (t, *J* = 6 Hz, 3H), 1.06 (s, 3H), 1.05 (s, 3H) ppm; ^13^C NMR (75.5 MHz, CDCl_3_) δ = 206.21, 199.22, 173.03, 157.97, 150.47, 127.26, 114.27, 79.08, 61.06, 59.93, 57.98, 45.66, 43.62, 42.09, 37.51, 34.07, 31.78, 30.36, 24.80, 14.36 ppm; IR (KBr):υ_max_ = 3374.71, 2985.31, 2963.32, 2945.34, 2928.06, 2873.40, 1726.97, 1705.28, 1670.93, 1633.80, 1616.71, 1228.42, 1203.67, 1085.27, 1045.22 cm^−1^; MS (ESI): *m*/*z*: 369; HRMS (ESI) Calcd. for [(C_20_H_26_O_5_) + Na]^+^: 369.1678; found: 369.1682 [M + Na]^+^.

#### 3.1.3. General Method for Synthesis of **16** and **18**

To a solution of **15** (1.0 eq) in EtOAc was added IBX (3.0 eq). The mixture was heated at 70 °C for 4 h and filtered through a short pad of silica gel washed with EtOAc. The solvent was removed under reduced pressure and the residue was purified by flash column chromatography to provide **16**.

Compound **16**: The compound is obtained as pale yellow liquid; ^1^H NMR (300 MHz, CDCl_3_) δ = 6.57 (d, *J* = 9 Hz, 1H), 6.19 (s, 1H), 5.83 (d, *J* = 9 Hz, 1H), 5.59 (s, 1H), 4.27 (q, *J* = 6 Hz, 2H), 3.68 (m, 2H), 2.77–2.68 (m, 4H), 2.11 (d, 1H), 1.85 (m, 2H), 1.29 (t, *J* = 6 Hz, 3H), 1.16 (s, 6H) ppm; ^13^C NMR (75.5 MHz, CDCl_3_) δ = 206.69, 198.72, 197.33, 168.85, 157.95, 142.16, 126.66, 120.53, 61.80, 61.16, 54.10, 46.06, 39.07, 37.43, 35.08, 33.73, 32.93, 30.59, 25.18, 14.19ppm; IR (KBr):υ_max_= 3434.29, 2962.17, 2929.37, 2871.25, 1722.07, 1675.43, 1638.37, 1249.74, 1177.04, 1111.95 cm^−1^; MS (ESI): *m*/*z*: 367; HRMS (ESI) Calcd. for [(C_20_H_24_O_5_) + Na]^+^: 367.1521; found: 367.1528 [M + Na]^+^.

Compound **18**: The compound is obtained as pale yellow liquid; ^1^H NMR (600 MHz, CDCl_3_) δ = 6.58 (d, *J* = 9.9 Hz, 1H), 6.13 (s, 1H), 5.78 (d, *J* = 9.9 Hz, 1H), 5.58 (s, 1H), 4.25 (q, *J* = 7.1 Hz, 2H), 3.78 (s, 1H), 2.93 (d, *J* = 12.2 Hz, 1H), 2.66 (m, 3H), 2.45 (m, 1H), 2.13 (d, *J* = 17.2 Hz, 1H), 1.87 (m, 2H), 1.31 (t, *J* = 7.1 Hz, 3H), 1.18 (s, 3H), 1.14 (s, 3H) ppm; ^13^C NMR (150 MHz, CDCl_3_) δ = 204.14, 200.19, 199.55, 169.71, 158.69, 142.90, 127.19, 119.93, 62.59, 61.80, 55.14, 45.85, 43.86, 40.88, 37.84, 34.18, 32.82, 30.26, 24.81, 14.18 ppm; IR (KBr):υ_max_ = 3435.06, 2961.46, 2931.17, 2871.34, 1739.98, 1720.04, 1671.03, 1638.37, 1241.36, 1228.03, 1177.31, 1142.38 cm^−1^; MS (ESI): *m*/*z*: 367; HRMS (ESI) Calcd. for [(C_20_H_24_O_5_) + Na]^+^: 367.1521; found: 367.1515 [M + Na]^+^.

### 3.2. Biological Evaluation

Cells were purchased from the Wuxi Innovatbio Medicine Technology Co. LTD (Wuxi, China) and were maintained at 37 °C under the atmosphere of 5% CO_2_. The cells of human non-small lung cancer (H292) and colon cancer cell line (SNU-1040) cells were cultured in RPMI-medium and others (rat myoblasts and HepG2) were cultured in dulbecco′s modified eagle medium (DMEM). 10% fetal bovine serum (Tianhang Biotechnology Co., Ltd., Zhejiang, China) and 1% antibiotics (100 U/mL penicillin and 100 mg/mL streptomycin) was supplemented. Cells are inoculated in 96-well plate and cultured for 24 h. After attached to the culture bottle wall cells were divided cells into three groups, (1) control group; (2) positive control group; (3) experiment group. The administration concentrations of Cisplatin (DTT) and compounds were 0.1 μM–10 μM. After 24 h, 10% MTT was added and cells were culture for 3.5 h. Then the culture medium was discard and 150 µL DMSO was added. Eventually, after incubation for 10 min by shaking in the dark at room temperature, the absorbance was detected at 490 nm using a microplate reader, and the difference was analyzed by prism 7 software.

## 4. Conclusions

In conclusion, we have designed and synthesized several compounds bearing two α,β-unsaturated ketone moiety. Their in vitro activities were evaluated against HepG2, NSCLC-H292, SUM-1040 tumor cell lines and an L6 myoblast cell line, with IC_50_ values ranging from 0.33 to 115.17 μM. This study has revealed that compounds **16**, **17** and **18** are promising anti-tumor leads due to the stronger cytotoxic potencies and higher selectivity they displayed than their natural counterparts.

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
