# Peer review of "Synthesis of Novel *ent*-Kaurane-Type Diterpenoid Derivatives Effective for Highly Aggressive Tumor Cells"

_molecules, 2018, doi:10.3390/molecules23123216_

Round 1
Reviewer 1 Report
Professor Liu and co-workers presented synthesis of Kaurane-type diterpenoid derivatives which show activities to cancer cell lines. Natural products, with electrophilic moieties, are of interest to synthetic chemists due to their potential reactivity with nucleophilic amino acid residues (cysteines, lysines, etc). A number of natural diterpenoids, with 1,2-unsaturated carbonyl structure were isolated and characterized, which show cytotoxic effects. However, many of them lack selectivity toward malignant cell lines, and are not suitable for future development. Therefore, development of derivatives with the key functionality are of importance. Liu's group presented synthesis a few analogues of diterpenoid natural products, and evaluated their biological activity against cancer cell lines. The synthetic routes were clearly described. The products were sufficiently characterized. The stereoisomers were unambiguously identified by X-ray crystallography. This reviewer has following suggestion:
Page 1, line 33: "commone", delete the letter "e"
Page 2, Figure 2: please label the number of atoms according to the paragraph above, so it will be more clear to the readers
Page 3, Scheme 1: Please write the full condition for the transformation from 11 to 12, even though it is a cited procedure.
Since there are multiple 1,2-unsaturated carbonyl moieties in the structure of compound 13-18, a more thorough study may be interesting to assess the necessity of each group (e.g. to reduce one of them to saturated alkanes each time)
Why the three human tumor cell lines were used to evaluate the activity of synthesized compounds? Any particular reason on molecular level?
This reviewer recommends acceptance after answering the questions above.
Author Response
1. Page 1, line 34, commone delete the letter “e” to common
2. Figure 2, labeled the number of atoms in the structure of eriocalyxin B
3. Page 3, scheme 1, write the full condition for the transformation from 11 to 12, 1. TBSOTf, Et3N, THF, -60℃; 2. 10% Pd(OAc)2, DMSO, O2, 40℃.
4. Thank you very much for your great efforts on our manuscript. As you said, the structure of dienone is necessary to maintain its anti-tumor activity. The effect of enone on its activity in the A or D ring has been reported in our previous studies (J. Heterocyclic Chem. 2012, 49, 571-575, DOI: 10.1002/jhet.816; Eur. J. Med. Chem. 2007, 42, 494-502, DOI: 10.1016/j.ejmech.2006.11.004). According to the literature (J. Med. Chem. 2017, 60, 1449−1468, DOI: 10.1021/acs.jmedchem.6b01652), unsaturated ketone in the D ring is the main active unit, while unsaturated ketone in the A ring only has weaker activity. This result is also consistent with the results tested in this paper. Because of the complex structure and low content of eriocalyxin B in plants, these restrict the application of eriocalyxin B in anti-cancer research. The purpose of this paper is to synthesize a number of compounds with dienone moeties like eriocalyxin B. At the same time, these compounds have simpler structures and better activities than their natural counterparts.
5. The three human tumor cell lines are the representative cells in the study of cancer. Non-small cell lung cancer is one of the most common malignant tumors in the world and has become the first cause of death of malignant tumors in China. Therefore, it is important to look for effective drugs to treat non-small cell lung cancer. Colon cancer cell line and HepG2 are both widely used in the research of anti-cancer drugs and a lot of article has reported.
For our compound, based on previous basic study we predict that it has an effect of anti-cancer, especially non-small cell lung cancer, colon cancer cell and HepG2. For these reasons, we choose to study the effect of compound on the three kinds of human tumor cell line. The underlying molecular mechanisms will be our future study directions.
Reviewer 2 Report
This manuscript reports synthetic process of six synthetic compounds and their cytotoxicities.
At least one of them including number of synthetic compounds and biological experiments should be performed more than the current status.
Recently, combinatorial synthesis can produce at least 20 to 30 derivatives based on the diverse substituents.
Also, if a few compounds show good cytotoxicities against cancer cell lines, to prove this result deep biological experiments such as flow cytometry, Western blot analysis, etc should be added.
Data about simple synthetic results and simple MTT assay can be published.
Author Response
Thank the reviewer for these precious comments and suggestions. We understand that flow cytometry (or Western blot analysis) may better reveal how the compounds induce cytotoxicity of cancer cells. However, in this study, we mainly focus on the in vitro activities against human tumor cell lines, and find which compounds have high selectivity between tumor cells and normal cell. We think that cytotoxic assessment may not be optimal, but it should be sufficient to conclude that experiments to simplify and modify eriocalyxin B have been successful. Moreover, it is difficult to complete a flow cytometry experiment in our laboratory within 10 days limited by the editor. We will conduct flow cytometry and Western blot analysis to understand detailed underlying cellular mechanisms of these lead compounds (e.g. inhibiting proliferation or inducing apoptosis, and signaling pathways leading to such changes, etc) in our next study.
Reviewer 3 Report
The synthesis methods are exactly described. There is no detailed description of experimental methods for cell cultures. The work contains abbreviations for the names of 3 commercial cell cultures (chapter 3.2). There is no explanation of the names of the lines used, description of the culture conditions of these cell lines and doses of added cytostatics. The results of cytotoxicity using the MTT test are not precisely described. The results can not be based on one applied MTT test.
Author Response
1. The full names of the four cell lines used have been explained below Table 1.
2. The specific steps of bioactivity screening experiments have been modified in this paper, as follows: Cells were purchased from the Wuxi Innovatbio Medicine Technology Co. LTD (Wuxi, China) and were maintained at 37℃ under the atmosphere of 5% CO2. The cells of human non-small lung cancer (H292) and colon cancer cell line (SNU-1040) cells were cultured in RPMI-medium and others (rat myoblasts and HepG2) were cultured in dulbecco's modified eagle medium (DMEM). 10% fetal bovine serum (Tianhang Biotechnology Co., Ltd., Zhejiang, China.) and 1% antibiotics (100 U/mL penicillin and 100 mg/mL streptomycin) was supplemented. Cells are inoculated in 96-well plate and cultured for 24 h. After attached to the culture bottle wall cells were divided cells into three groups, 1) control group; 2) positive control group; 3) experiment group. The administration concentrations of Cisplatin (DTT) and compounds were 0.1μM-10 μmol/L. After 24 h, 10% MTT was added and cells were culture for 3.5 h. Then the culture medium was discard and 150 μL DMSO was added. Eventually, after incubation for 10 min by shaking in the dark at room temperature, the absorbance was detected at 490 nm using a microplate reader, and the difference was analyzed by prism 7 software.
Round 2
Reviewer 1 Report
The author answered the questions well. This reviewer agrees to publish the manuscript in the present form.
Reviewer 2 Report
.
Reviewer 3 Report
Interesting review work on a very important issue, which is the therapy of various cancers.